# Seasonal variation in species composition, deltamethrin susceptibility, and *kdr* mutations in anopheles mosquitoes in Northwest Ethiopia

Ligabaw Worku[1]*, Amha Kebede[2], Ayalew Jejaw Zeleke[1], Saron Fekadu[2,3], Melat Abdo[2], Tigist Atele[2], Netsanet Worku[4], Mulugeta Aemero[1]

**1** Department of Medical Parasitology, School of Biomedical and Laboratory Sciences, College of Medicine and Health Sciences, University of Gondar, Gondar, Ethiopia, **2** Armauer Hansen Research Institute, Addis Ababa, Ethiopia, **3** Department of Microbial Science and Genetics, College of Computational and Natural Sciences, Addis Ababa University, Addis Ababa, Ethiopia, **4** Institute of Public Health, College of Medicine and Health Sciences, University of Gondar, Gondar, Ethiopia

* ligabaw@gmail.com

## Abstract

### Background

Anopheles mosquitoes are the main vectors of malaria. Effective vector control depends on understanding their species composition, behavior, distribution, and insecticide resistance. This study investigated Anopheles species composition, susceptibility to deltamethrin, and the frequency of knockdown resistance (*kdr*) mutations in Maksegnit and Gendawuha, Northwest Ethiopia.

### Methods

Anopheles larvae and pupae were collected from breeding sites during the rainy and post-rainy seasons and reared to adults under field insectary conditions following WHO guidelines. In addition, adult mosquitoes were collected from houses near larval habitats. Only field-derived mosquito populations were used in this study. Adult females (3–5 days old) reared from field-collected larvae were tested for susceptibility to 0.05% deltamethrin using WHO bioassays. Based on bioassay outcomes, mosquitoes were classified as phenotypically susceptible (died after exposure) or resistant (survived exposure), while field-collected adults represented an unexposed group. A total of 480 mosquitoes (160 resistant, 160 susceptible, and 160 field-collected unexposed adults) were subjected to genomic DNA extraction. Species identification and detection of knockdown resistance (*kdr*) mutations (L1014F and L1014S) were performed using PCR.

### Results

WHO bioassays conducted on 776 mosquitoes revealed confirmed resistance to deltamethrin, with mortality rates ranging from 48.5% to 72.5% (overall resistance: 37.5%). Resistance intensity exhibited significant variation, peaking after the rainy

**Data availability statement:** All relevant data are within the manuscript and its Supporting Information files.

**Funding:** The author(s) received no specific funding for this work.

**Competing interests:** The authors have declared that no competing interests exist.

season and showing a higher prevalence in Maksegnit compared to Gendawuha (p < 0.05). Molecular identification of 480 mosquitoes showed that *Anopheles arabiensis* was the predominant species (93%, 446/480), followed by *An. pharoensis* (6%, 29/480) and *An. stephensi* (1%, z/480), with the latter detected for the first time in Gendawuha. Regarding kdr mutation status, genotypic analysis showed that the L1014F mutation was the predominant allele, particularly among phenotypically resistant mosquitoes (67.8%), while lower frequencies were observed in susceptible (45.8%) and unexposed field-collected groups (61.4%). Conversely, the L1014S mutation was detected at low frequency (≤12.3%) and was restricted exclusively to the Maksegnit population.

## Conclusion

*Anopheles arabiensis* predominated, with confirmed resistance to deltamethrin, particularly in the post-rainy season. The L1014F *kdr* mutation was prevalent, while L1014S *kdr* mutation was rare. Detection of *Anopheles stephensi* highlights emerging risks, underscoring the need for season-specific resistance monitoring and integrated control strategies.

---

## Background

Anopheles mosquitoes are the main vectors of malaria globally, transmitting Plasmodium parasites that lead to high morbidity and mortality, particularly in sub-Saharan Africa where the disease burden is greatest [1]. Understanding their biology, behavior, and distribution is critical for designing targeted vector control strategies aimed at reducing transmission [2]. Effective management of these mosquito populations remains a significant public health challenge, emphasizing the need for ongoing surveillance, resistance monitoring, and integrated control approaches [3]. Knowledge of species composition, seasonal patterns, and geographical distribution is essential for implementing precise interventions tailored to local vector ecology, such as insecticide application and habitat modification, timed to peak mosquito activity, and focused in high-risk areas to minimize transmission and enhance public health outcomes [4–8].

Insecticide resistance poses a significant challenge to malaria control efforts, with knockdown resistance (*kdr*) mutations playing a key role in this phenomenon [9]. The L1014F and L1014S alleles are specific mutations in the voltage-gated sodium channel gene of Anopheles mosquitoes that reduce their sensitivity to pyrethroids like deltamethrin, commonly used in insecticide-treated nets and indoor residual spraying [10]..These mutations change the insecticide target site, making the insecticide less effective. As a result, mosquitoes can survive exposure to WHO diagnostic dose, thereby compromising control measures and potentially leading to increased malaria transmission [10].

A distinct mutation was identified in eastern Sudan (Khartoum State) and in Juba and Wau, South Sudan, involving a substitution from leucine (TTA) to serine (TCA) at the same amino acid position associated with the *kdr* L1014S allele [11,12]. More

recently, *Anopheles arabiensis* [5,13,14] and *Anopheles stephensi* from Eastern Ethiopia [15] were found to carry a leucine (TTA) to phenylalanine (TTT) mutation. Likewise, mosquito samples from Kenya revealed the leucine (TTA) to serine (TCA) substitution [16].

Existing knockdown resistance (*kdr*) mutations, notably L1014F and L1014S alleles, are increasingly prevalent in Ethiopian and broader African Anopheles mosquito populations, contributing to varying resistance patterns against pyrethroids like deltamethrin [17]. In Ethiopia, studies have documented the presence of both mutations, with L1014F being more common in certain regions, indicating localized selection pressure due to extensive insecticide use [13,18]. Across Africa, the distribution of these alleles varies geographically, often correlating with documented resistance in vector populations, which complicates control efforts [5,19–24]. The widespread presence of *kdr* mutations underscores the need for resistance management strategies, including rotation of insecticides and integrated vector control approaches, to sustain the efficacy of malaria interventions.

Despite these findings, we still lack updated information on the current species composition, seasonality, and geographical variation of Anopheles mosquitoes in the region. Furthermore, neither the L1014F nor the L1014S allele mutations have been reported in Northwest Ethiopia to date. This study seeks to assess the species composition, deltamethrin susceptibility, and *kdr* mutations in Anopheles mosquito populations across two transmission seasons in Maksegnit and Gendawuha towns of Northwest Ethiopia. It will examine the species composition, seasonal and geographical variation of Anopheles mosquito populations, and the prevalence and allele frequency of the *kdr* mutations. The results are intended to guide and optimize vector management strategies in the region, ensuring more effective malaria control interventions.

## Methods and materials

### Study areas

The study was conducted between June and December 2023 in two districts of northwest Ethiopia to capture seasonal variations in Anopheles mosquito species composition and deltamethrin resistance status (Fig 1). The first site, Gondar Zuria district, Maksegnit Town, situated at 1,923 meters above sea level in the Central Gondar Zone, experiences cooler temperatures (14–20°C) and receives 1030–1223 mm of annual rainfall, reflecting low malaria transmission zones. The second site, Gendawuha Town in the Metema district of the West Gondar Zone, lies at 685 meters elevation, with higher temperatures (22–28°C), peak temperatures reaching 43°C from March to May, and annual rainfall of 850–1000 mm, characteristic of high transmission areas. Malaria control efforts include periodic indoor residual spraying (IRS) since 2011 using carbamates and the distribution of pyrethroid-treated long-lasting insecticidal nets (LLINs) every three years, aligned with family sizes. Malaria transmission is seasonal, peaking from September to December, with reduced levels from April to July, underscoring the importance of understanding vector dynamics and insecticide resistance in these varying ecological contexts.

### Ethical clearance

Ethical approval was granted by the University of Gondar's Institutional Review Board (Ref. No. 06/ 314/2015 E. C). Field site access and mosquito collections were authorized by the respective District Health Offices. Following an explanation of the study's purpose in the local language, household heads provided verbal informed consent a procedure specifically approved by the Ethics Committee and documented in a field log.

### Mosquito larvae, pupa collection, rearing, and adult collection

Dipping, as a larval sampling technique, was conducted during the rainy and post-rainy seasons following World Health Organization (WHO) guidelines and standard operating procedures for entomological surveillance [25] for Anopheles larvae and pupae collection. Collected larvae and pupae were reared in field insectaries set up at each study location, using water drawn from their natural breeding habitats. Approximately 100–150 larvae were maintained per rearing tray,

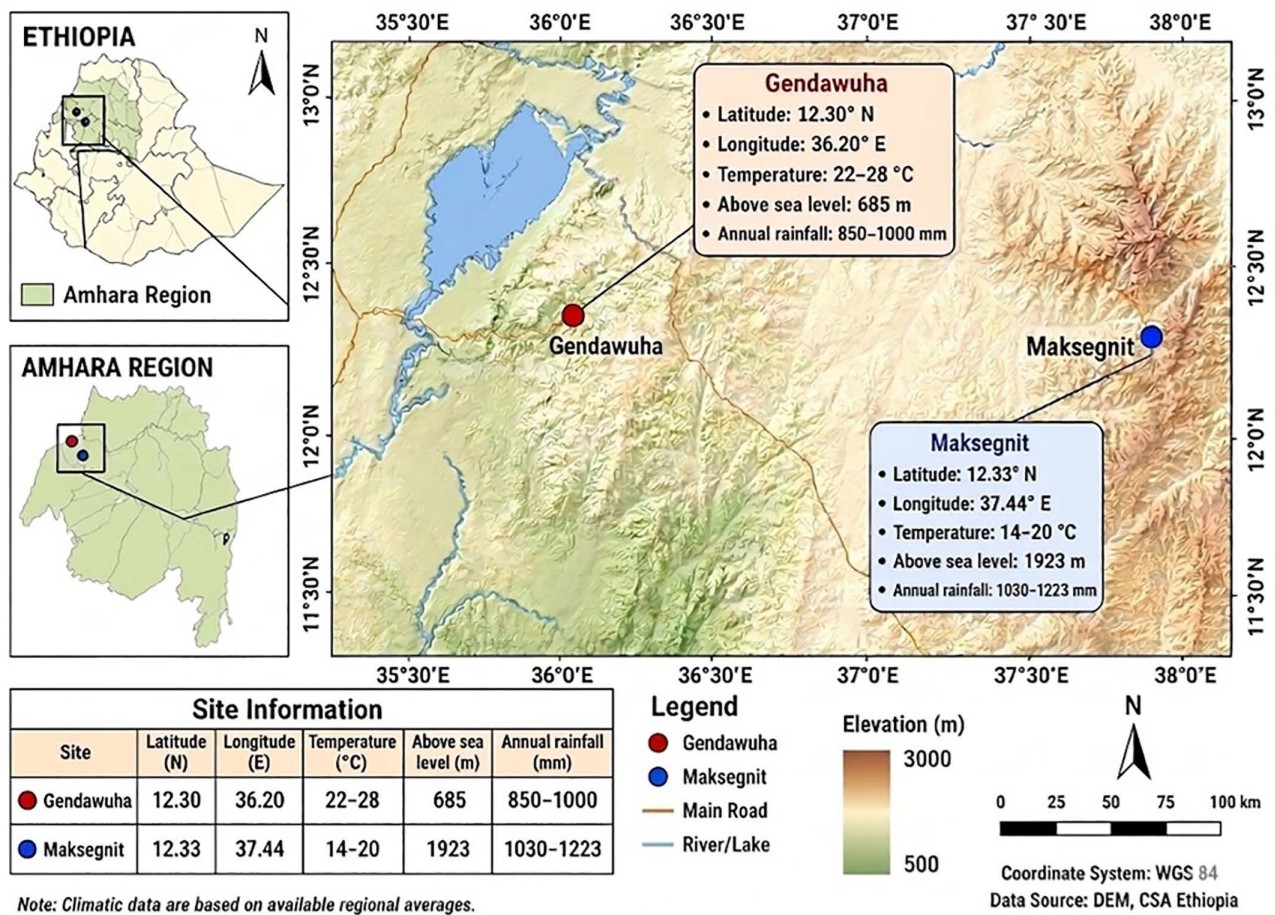

**Fig 1. Map of the study area in Northwest Ethiopia.** The map illustrates the geographical locations of the study sites, Gendawuha (red dot) and Maksegnit (blue dot), situated within the Amhara Region. Insets show the national context within Ethiopia and regional context within the Amhara Region. The elevation profile, major roads, and water bodies are indicated, with site-specific climatic and topographical data including latitude, longitude, temperature, elevation, and annual rainfall summarized in the accompanying table and callout boxes.

and they were fed daily with about 0.02–0.05 g of Tetramin fish food (Tetramin Tropical Flakes, Spectrum Brands, Inc.) per tray. The larvae were exposed to natural sunlight during the daytime (S1 Fig). Pupae were collected each morning using a pipette and transferred into adult emergence cages, where they were allowed to emerge into adults. (S2 Fig). Adult female Anopheles mosquitoes were tested for insecticide susceptibility using the WHO bio-assay (S3A Fig), and their sibling species were determined through species-specific polymerase chain reaction (PCR) assays [26]. Adult female Anopheles mosquitoes were also collected in the house near the larva and pupa collection site using Prokopack for species identification using molecular methods (S3B Fig).

## Insecticide susceptibility test

Insecticide susceptibility tests were carried out following the WHO insecticide susceptibility test procedure [27]. Twenty-five 3–5-day-old female Anopheles mosquitoes from field larvae collection, previously fed a 10% sugar solution, were exposed to 0.05% deltamethrin-impregnated papers for one hour in WHO test tubes. Control groups were exposed to untreated papers under the same conditions. After one-hour of exposure, the mosquitoes were transferred to holding

tubes with a 10% sucrose solution, and mortality was recorded after 24 hours. When control mortality was between 5% and 20%, treated mortality were corrected using Abbott's formula [28], as recommended by WHO guidelines. Following the bio-assay, all alive and dead specimens were maintained separately in Eppendorf tubes in 95% alcohol for molecular identification and *kdr* PCR testing. Resistance was determined based on WHO criteria as follows: 98–100% mortality indicates susceptibility, 90–97% mortality indicates a resistance candidate, and less than 90% mortality suggests resistance [27].

## DNA extraction and species identification

Genomic DNA was extracted from a total of 480 individual mosquito (160 survived and 160 died from the bio-assay tested, and 160 were adult collected) sampled from Maksegnit and Gendawuha using the method of Cetyl Trimethyl Ammonium Bromide (CTAB) [29]. Tissue samples from the legs and wings were homogenized with silica beads and molecular-grade water in a Mini-Bead Beater 16 (BioSpec Products, Bartlesville, OK, USA). The homogenate was then incubated with CTAB buffer and chloroform to facilitate DNA separation, followed by centrifugation using a refrigerated micro centrifuge. The supernatant was treated with isopropanol to precipitate DNA, which was then washed with ethanol, dried, and re-suspended in PCR-grade water. The extracted DNA was stored at 4°C for short-term and at −20°C for long-term preservation.

The specimens were characterized into sibling species using PCR [30]. PCR amplification was performed using a T100 thermal cycler (Bio-Rad, Hercules, CA, USA) using Species-specific primers (primer sequence 5' to 3'): universal (GTGT-GCCCCTTCCTCGAT GT), *An. gambiae s.s.* (CTGGTTTGGTCGGCACGTTT), *An. merus* and *An. melas* (TGACCA ACCCACTCCCTTGA), *An. arabiensis* (AAGTGTCCTTCTCCATCCTA), and *An. quadriannulatus* (CAGACCAAGAT-GGTTAGTAT) designed from the DNA sequence of the intergenic spacer region of *An. gambiae s.l.* were used for the identification [31]. Polymerase chain reaction amplification was carried out with an initial denaturation step at 95°C for 5 min., followed by 30 cycles each consisting of 30 seconds of denaturation at 95°C, 30 seconds of annealing at 50°C, and 30 seconds of elongation at 72°C. The final elongation was carried out at 72°C for 5 minutes. PCR products negative for *An. gambiae* complex groups were further identified for *An. stephensi:* st-F (CGTATCTTTCCTCGCATCCA) and UD2-R (GCACTA TCAAGCAACACGACT), and *An. pharoensis* (TCTAATATGGGAGATTAGTGC and ACTT GCTTTCAGT-CATCTAATG) were used as forward and reverse primers, respectively. The PCR products were electrophoresed through ethidium bromide-stained 2% agarose gel using a gel electrophoresis unit (Bio-Rad, USA) and visualized under UV light in an E-Box CX5 gel documentation system (Vilber, France). Species identification was determined based on the presence or absence of diagnostic DNA bands at expected molecular weights. Each gel lane corresponding to an individual mosquito specimen was independently scored by two researchers using a binary scoring approach. Ambiguous or faint bands were resolved using the image enhancement tools of the gel documentation software.

## Detection of the L1014F and L1014S *kdr* alleles

Allele-specific PCR assay was conducted on the same mosquito samples used for the identification of species. The presence of L1014F and L1014S *kdr* alleles' mutation was detected by adapting the established protocols [32,33]. The PCR was conducted in a 20 μl reaction volume using species- and allele-specific primers in a T100 thermal cycler (Bio-Rad, Hercules, CA, USA). Amplification followed a program of initial denaturation at 94°C, followed by 40 cycles of denaturation, annealing at 48°C, extension, and a final extension at 72°C. PCR products were separated on a 2% agarose gel using a gel electrophoresis system (Bio-Rad, USA) and visualized under UV illumination using an E-Box CX5 gel documentation system (Vilber, France). Genotypes were classified as homozygous resistant (RR), heterozygous (RS), or susceptible (SS) based on the presence or absence of allele-specific banding patterns. Each gel image was independently scored by two researchers, and unclear bands were verified using the contrast enhancement tools of the gel documentation software before genotype assignment.

## Data analysis

We used the WHO 2016 classification standards to analyze the data on susceptibility status: 24- hour mortality rate of 98% or higher (Susceptible); a mortality rate of 90–98% (Suspected); and a mortality rate of less than 90% (Resistance). We used Abbott's formula to adjust the average observed mortality when the control mortality was between 5% and 20% [28]. When the control mortality exceeded 20%, the test was repeated. Finally, the mortality rates, genotypes, and *kdr* allelic frequencies for each studied population were determined. All statistical analyses, tables, pie charts and graphs were produced using Microsoft Excel and SPSS, version 27.0.

## Results

### Species composition

A total of 480 Anopheles mosquitoes were analyzed in this study. This included 360 individuals primarily identified through WHO susceptibility testing for deltamethrin and 120 wild-collected adults specifically utilized for baseline molecular species identification (Fig 2B-D). Overall molecular identification confirmed that *An. arabiensis* was the predominant species, representing 93% (n = 447) of the total, followed by *An. pharoensis* 6% (n = 30) and *An. stephensi* 1% (n = 3) (Fig 2A). Statistical analysis revealed no significant seasonal variation in species distribution (p = 0.982). *Anopheles arabiensis* remained the dominant vector across all study sites, seasons, and experimental groups, including both the susceptibility-tested and the baseline molecular cohorts (Fig 2B-D). While geographic variation was evident through the first molecular confirmation of *An. stephensi* in Gendawuha, the overall species composition did not differ significantly between Gendawuha and Maksegnit (p > 0.05).

### Deltamethrin susceptibility

WHO tube bio-assays revealed varying mortality rates following 0.05% deltamethrin exposure, ranging from 48.5% to 72.5% after 24 hours. These differences were significant across sites and seasons (p < 0.05). According to WHO thresholds, confirmed resistance (< 90% mortality) was observed in populations from both Maksegnit and Gendawuha. Resistance tended to be higher after the rainy season compared to rainy season, when vector densities were greatest. Geographic variation was also evident, with mosquitoes from Maksegnit showing stronger resistance compared to those from Gendawuha (Table 1).

### Detection of L1014F and L1014S *kdr* alleles

To detect the presence of the knockdown resistance (*kdr*) alleles L1014F and L1014S, Allele-Specific PCR (AS-PCR) was employed in a total of 480 Anopheles mosquito samples. Of these, 465 specimens (96.9%) were successfully genotype for the L1014F allele and 462 specimens (96.3%) for the L1014S allele. To analyze the relationship between genotype and resistance phenotype, mosquitoes were grouped into three categories: field-collected adults (FCA, n = 160) which were the unexposed individuals collected from houses near the larval collection site using a different method than the mosquitoes subsequently used for bioassays, phenotypically susceptible mosquitoes (n = 160) which died after exposure, and phenotypically resistant mosquitoes (exposure = 160). Analysis of the *kdr* mutation prevalence revealed that the L1014F mutation is the dominant genotype, exhibiting significantly higher prevalence across all populations compared to the L1014S allele. Its highest frequency was observed in the phenotypically resistant group at 67.8%. The field-collected adults group showed a high L1014F prevalence of 61.4% and notably hosted the highest rate of the less common L1014S mutation, at 12.3%. Conversely, the phenotypically susceptible group displayed the lowest prevalence for both mutations, with L1014F at 45.8% and L1014S at just 5.6% (Fig 3).

The frequency of *kdr* mutations (L1014F and L1014S) varied by site and season (Fig 4A–B). The L1014F mutation was more prevalent in Maksegnit than in Gendawuha, with higher frequencies of homozygous resistant genotypes (RR)

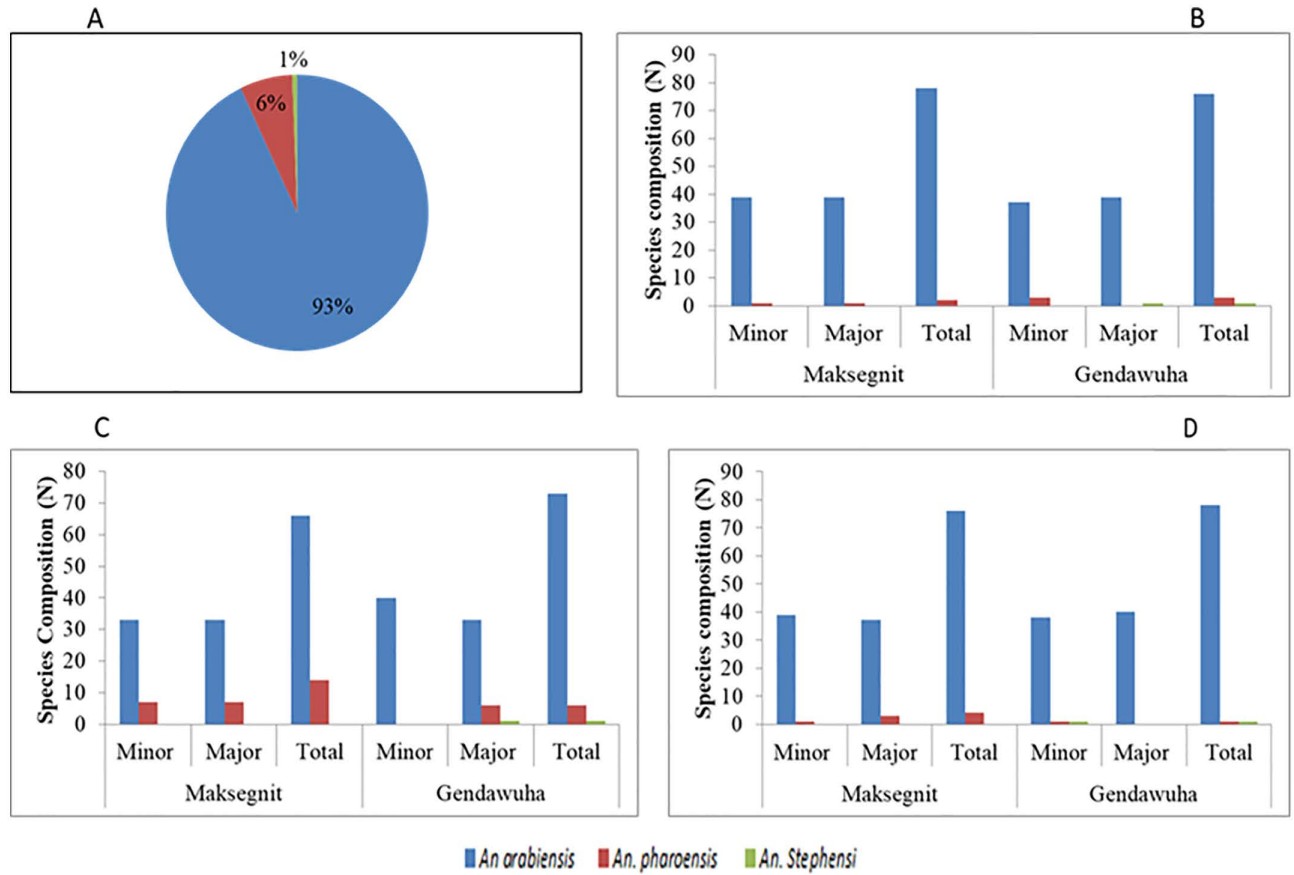

**Fig 2. (A)** Prevalence of mosquito species **(B)** Species composition frequency by season and site for field collected adult **(C)**, phenotypically susceptible mosquitoes after exposure to deltamethrin species frequency **(D)**, phenotypically resistant mosquitoes after exposure to deltamethrin species frequency.

**Table 1. Susceptibility status of Anopheles mosquitoes to 0.05% deltamethrin across two seasons in Maksegnit and Gendawuha, Northwest Ethiopia.**

| Study sites | Seasons | No of exposed mosquitos | Susceptible/ died | Resistance/live | χ2 | P – value |
|---|---|---|---|---|---|---|
| Maksegnit | Rainy | 184 | 123(67%) | 61(33%) | 13.185 | 0.0002 |
| | After rainy | 200 | 97(48.5%) | 103(51.5%) | | |
| Gendawuha | Rainy | 200 | 144(72.5%) | 55(27.5%) | 4.12 | 0.042 |
| | After rainy | 192 | 121(63%) | 72(37%) | | |
| Total | | 776 | 485(62.5%) | 291(37.5%) | | |

Key: – Rainy season (June to August); after rainy season (September to December).

observed following the rainy season, suggesting stronger selection pressure. In contrast, Gendawuha populations were largely homozygous susceptible (SS), indicating partial recovery of susceptibility. The L1014S mutation occurred at much lower frequencies, detected only in Maksegnit, where it was more common during the rainy season. No L1014S alleles were detected in Gendawuha across either season.

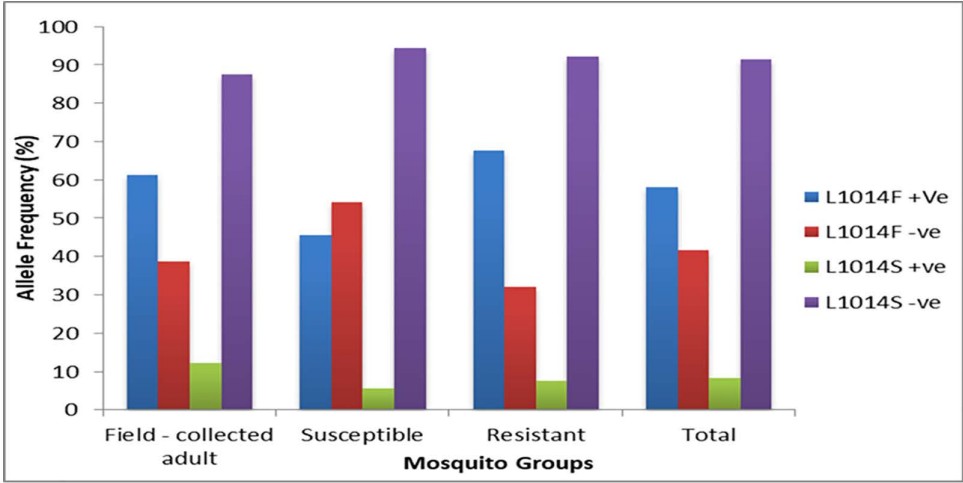

**Fig 3. Distribution of L1014F and L1014S *kdr* allele frequencies among field-collected, phenotypically susceptible, and phenotypically resistant Anopheles mosquito groups in Northwest Ethiopia.**

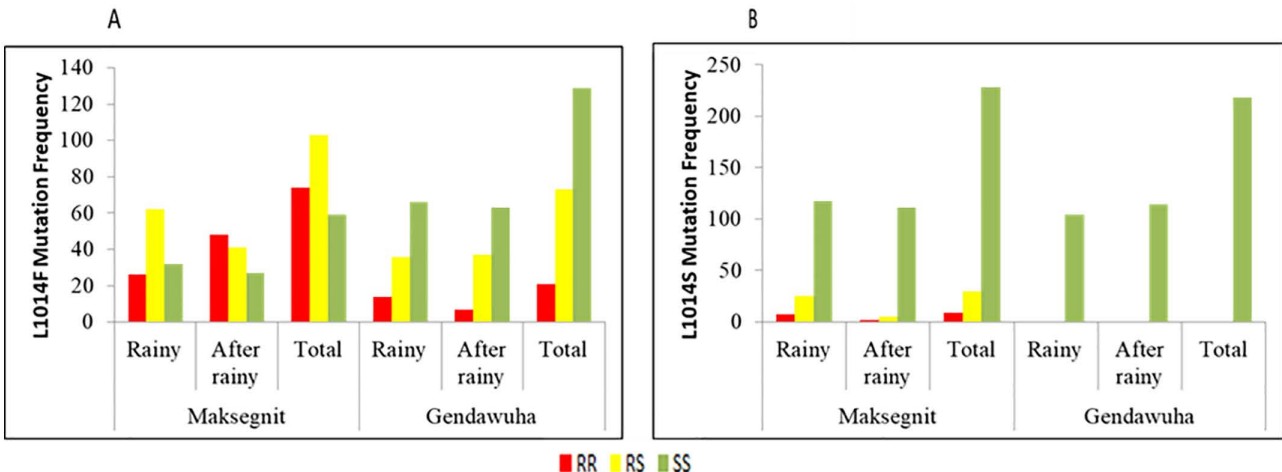

**Fig 4. Distribution and frequency of knockdown resistance (*kdr*) mutations L1014F (A) and L1014S (B) in *Anopheles* mosquitoes across different seasons and study sites.** The bar charts illustrate the frequency of genotypes (RR: Homozygous Resistant; RS: Heterozygous; SS: Homozygous Susceptible) in Maksegnit and Gendawuha during the Rainy and After-rainy seasons.

When we examined both field-collected adults and susceptible *An. arabiensis* for the *kdr* L1014F allele mutant gene showed a significant number of heterozygous genotype across all study sites and seasons. However, among the *An. arabiensis* identified as resistant by the WHO bio-assay, there was a notably high number of homozygous genotype. Similar trend for the *kdr* L1014S allele mutation was also observed (Fig 5).

## Discussion

Effective malaria control depends on detailed knowledge of Anopheles species composition, seasonal fluctuations, and geographical distribution, particularly in northwest Ethiopia, where such information guides targeted vector control

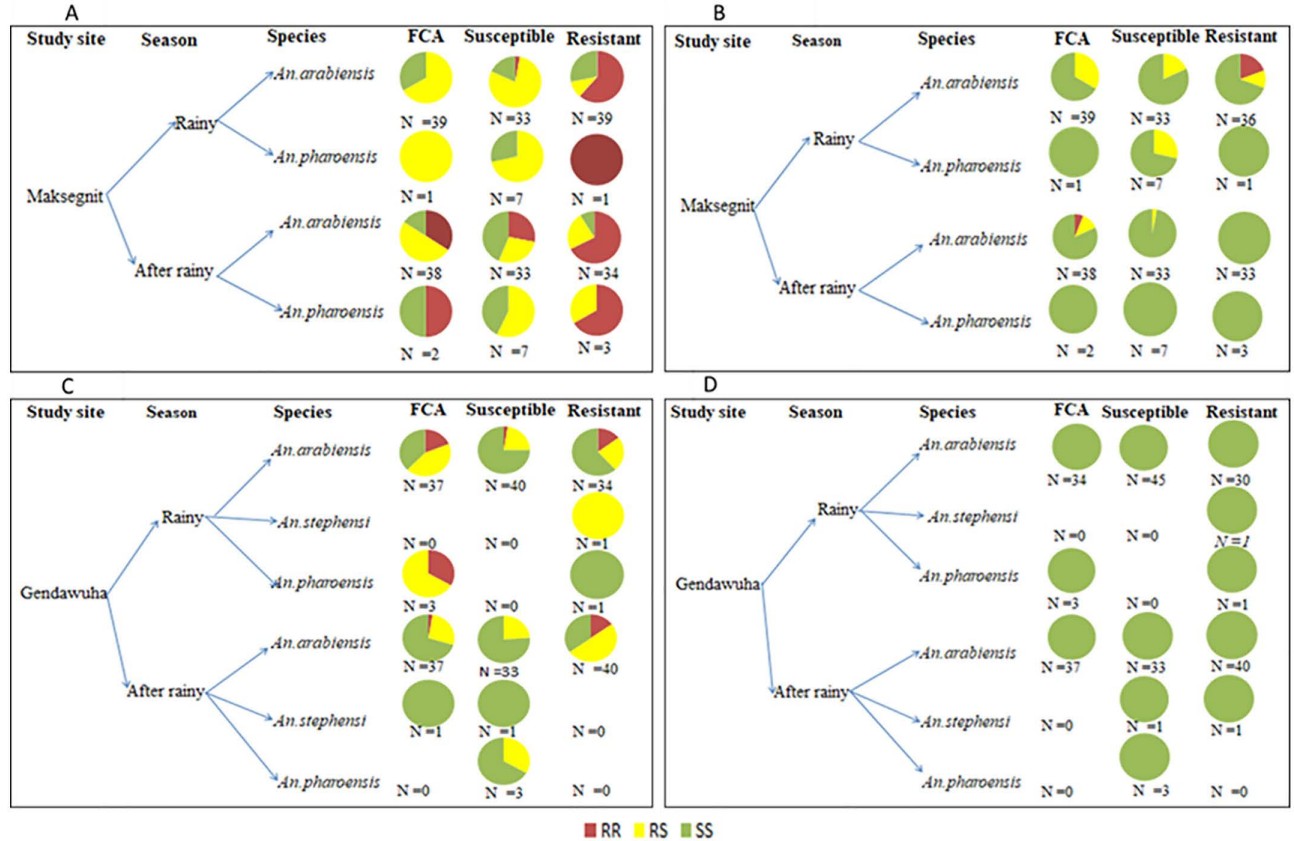

**Fig 5. Distribution of *kdr* genotypes in *Anopheles* species from Northwest Ethiopia.** The figure displays the frequency of L1014F and L1014S genotypes RR (resistant), RS (heterozygous), and SS (susceptible) across Maksegnit and Gendawuha. Data is stratified by Rainy and After-rainy seasons for *An. arabiensis*, *An. pharoensis*, and *An. stephensi*. Comparisons are made between field-collected adults (FCA) and those categorized as Susceptible or Resistant after insecticide exposure. Sample sizes (N) are provided for each group.

interventions. The spread of resistance alleles (*kdr* L1014F and L1014S) further challenges pyrethroid-based tools such as ITNs and IRS [23].

Our findings show that *An. arabiensis* was the dominant species, comprising ~93% of all specimens across sites, seasons, and mosquito groups (range 82.5–100%). This agrees with earlier findings identifying *An. arabiensis* as the main malaria vector in northwest Ethiopia, followed by *An. pharoensis* (6%) and *An. stephensi* (1%) [5,34]. The presence of *An. stephensi* in Gendawuha near the Sudanese border indicates its continued geographical expansion into northwestern Ethiopia, likely from established populations eastern Ethiopia or Sudan [35–42]. This presence is particularly concerning given that *An. stephensi*, which is native to South and Southeast Asia [43,44], is a highly efficient urban vector known for its capacity to thrive in artificial human-made habitats such as water storage containers, tanks, and drainage systems, posing a severe and emerging public health threat [45].

The establishment of *An. stephensi* in Ethiopia represents a significant emerging threat, as it can sustain urban malaria transmission and weaken control strategies traditionally aimed at rural vectors [43]. Its spread is likely being accelerated by human activities such as trade and transportation, which may inadvertently move larvae or adult mosquitoes between regions [46]. This adaptability emphasizes the need for targeted measures, including urban larval source management (LSM) focused on water tanks and discarded containers, complemented by IRS, ITN distribution, and improved

community water-storage practices [47]. The detection of *An. stephensi* in northwest Ethiopia also highlights the necessity of strengthened entomological surveillance, ongoing monitoring of species distribution and insecticide resistance, and integrated vector management that addresses both dominant rural vectors like *An. arabiensis* and emerging urban species.

This study also evaluated deltamethrin susceptibility and *kdr* allele detection in *Anopheles* mosquitoes. We collected larvae from natural breeding sites and then reared them to obtain adult females for bioassays. The results showed a substantial level of deltamethrin resistance (~ 63% mortality). This finding aligns with established patterns in Ethiopia [5,14,24] and suggests a slight increase compared to earlier data from southwestern and other regions [24,48,49]. However, this resistance level is still lower than the high levels documented in central Ethiopia, Arjo Dedessa, and the Lake Tana region, where resistance in *An. arabiensis* is particularly well-established [5,14,50].

Regional differences in deltamethrin resistance likely reflects interacting factors, including insecticide use patterns, control strategies, ecological conditions, and vector genetics. Intensive exposure through IRS or widespread ITN use increase selection pressure and accelerates resistance alleles spread [9,51]. Environmental factors such as breeding site diversity and species composition also shape resistance development [52].

Resistance peaked after the rainy season (September–December), reaching 52% in Maksegnit and 37% in Gendawuha. This rise likely reflects intensified insecticide use during peak transmission and higher mosquito densities that promote the survival and spread of resistant phenotypes [23]. These findings highlight the dynamic nature of resistance evolution and the need for continuous, site-specific monitoring to inform adaptive and evidence-based vector control strategies.

Molecular analysis of Anopheles from Maksegnit and Gendawuha confirmed widespread *kdr* mutations (L1014F and L1014S), with clear spatial and seasonal variation. Using allele-specific PCR, more than 92% of samples were successfully genotyped, revealing the dominance of the L1014F mutation, particularly during the rainy season in Maksegnit, where both RR and RS genotype were common. The L1014S allele, historically associated with East African vector populations, was also detected and occurred more frequently in Maksegnit than in Gendawuha, indicating local differentiation in allele distribution [13].

Mosquitoes were categorized as wild-collected adults, phenotypically susceptible, or phenotypically resistant to evaluate genotype–phenotype links. Analysis of *kdr* mutation prevalence across these groups further solidified the functional importance of the L1014F allele. The L1014F mutation was the dominant genotype in all categories, with its frequency peaking significantly in the phenotypically resistant group. This strong genotypic association provides definitive evidence of the functional link between the L1014F allele and pyrethroid resistance in the tested population [53]. Conversely, the phenotypically susceptible group displayed the lowest prevalence for both mutations (L1014F and L1014S), confirming that mosquitoes lacking these mutant alleles are less likely to survive deltamethrin exposure [54,55] Notably, the field-collected adults group showed a high baseline L1014F prevalence and hosted the highest rate of the less common L1014S mutation [20]. The high L1014F frequency in unexposed field- collected mosquitoes suggests widespread and sustained selection pressure on the vector population in the natural environment, likely due to intensive use of pyrethroids in ITNs and IRS across the study area [56,57].

The predominance of heterozygous genotypes among field-collected adults and insecticide-susceptible *An. arabiensis*, alongside a higher proportion of homozygous resistant individuals among bio-assay survivors, suggests active selection and a functional role of these alleles in resistance [58]. The coexistence and spatiotemporal variation of multiple *kdr* alleles further highlight the complexity of resistance mechanisms and the adaptive capacity of mosquito populations under sustained insecticide pressure [23].

These findings are consistent with reports from Ethiopia and other parts of sub-Saharan Africa, where intensive use of pyrethroids and DDT has driven a rising in *kdr* mutation frequency. In southwestern Ethiopia the L1014F allele nearly fixed in *An. arabiensis* and *An. gambiae* populations, often exceeding 80–90% allele frequency and strongly linked to pyrethroid

resistance [14,53,58]. In contrast, Uganda, Kenya, and Tanzania report higher L1014S allele frequency, reflecting regional differences in insecticide selection pressures [55,58–60] while its occurrence in Sudan suggests cross-border gene flow (55,56). In West Africa, both mutations are found, although L1014F remains predominant, influenced by local insecticide use patterns and ecological variation [20,61,62].

Ethiopia's resistance landscape is complex, characterized by the co-occurrence of both L1014F and L1014S mutations, similar to patterns observed in Kenya, and indicative of multiple, overlapping resistance pathways [13,58,63]. Seasonal increase in allele frequencies, following intensive insecticide use, demonstrates the evolutionary responsiveness of Anopheles populations [23]. Elevated heterozygosity further indicates ongoing gene flow and active selection dynamics.

These findings reveal a continent-wide trend of escalating pyrethroid resistance, undermining ITNs and IRS. Effective malaria control will depend on integrated vector management that combines molecular surveillance, insecticide rotation, and new tools [64,65]. Continuous genetic monitoring is essential to detect emerging resistance patterns and support evidence-based decision-making for future malaria control programs.

The limitations of the study are as follows: Study scope is limited to two sites in northwest Ethiopia, potentially under-representing regional variability. Sampling from June to December may miss seasonal or long-term trends. Molecular methods may not capture all resistance mechanisms, such as metabolic resistance. The cross-sectional design precludes causal inference, and low-frequency alleles or species such as *An. Stephensi* may be under detected due to sampling or technical limitations.

## Conclusion

This study reveals an evolving species composition dominated by *An. arabiensis*, alongside the first molecular confirmation of the invasive species *An. stephensi*, raising serious concerns about urban malaria transmission. Seasonal variation plays a significant role, with insecticide resistance patterns marked by high frequencies of *kdr* L1014F and L1014S alleles, particularly during peak transmission periods following the rainy season, which compromises the effectiveness of current interventions like IRS and LLINs. The observed resistance patterns, including heterozygous and homozygous resistant genotype, suggest active selection pressures driven by extensive insecticide use, further complicating control efforts in the region.

Based on these findings, policymakers should prioritize insecticide rotations and consider innovative tools such as spatial repellents. Future research should include longitudinal studies to monitor resistance trends over time, and explore the genetic mechanisms underlying resistance. Targeted studies of the invasive *An. stephensi* and urban transmission dynamics are also essential to adapt control strategies.

## Supporting information

**S1 Fig. Collected larvae and pupae were reared in field insectaries established at the study sites.**
(TIF)

**S2 Fig. Emergence of adult Anopheles mosquitoes from pupae in field insectaries.** The figure represent the final stages of the rearing process, where field-collected pupae transition into adults within controlled cages.
(TIF)

**S3 Fig. Insecticide susceptibility testing and indoor adult collection. A:** Displays the WHO susceptibility bio-assay tubes used to assess phenotypic resistance. **B:** Shows indoor adult female collection using a Prokopack aspirator in houses near larval breeding sites.
(TIF)

**S4 Fig. Graphical abstract.**
(TIF)

**S1 Data. Raw dataset for mosquito species and *kdr* genotypes.** Underlying data for all reported findings, including site-specific collection records, species composition, and the distribution of knockdown resistance (*kdr*) alleles among different mosquito groups.
(XLSX)

**S2 Data. Summary data and values used to generate manuscript figures.** Spreadsheet containing the processed values, frequencies, and totals used to build Figs 2, 3, and 4.
(XLSX)

## Acknowledgments

The authors would like to thank the Department of Medical Parasitology, University of Gondar; the Armauer Hansen Research Institute (AHRI); and technical staff members of AHRI, Ethiopia

## Author contributions

**Conceptualization:** Ligabaw Worku, Mulugeta Aemero.

**Data curation:** Ligabaw Worku, Saron Fekadu.

**Formal analysis:** Ligabaw Worku, Ayalew Jejaw Zeleke, Netsanet Worku, Mulugeta Aemero.

**Investigation:** Ligabaw Worku.

**Methodology:** Ligabaw Worku, Saron Fekadu, Melat Abdo, Tigist Atele.

**Supervision:** Amha Kebede, Netsanet Worku, Mulugeta Aemero.

**Validation:** Melat Abdo.

**Visualization:** Ligabaw Worku, Netsanet Worku.

**Writing – original draft:** Ligabaw Worku.

**Writing – review & editing:** Amha Kebede, Ayalew Jejaw Zeleke, Saron Fekadu, Melat Abdo, Tigist Atele, Netsanet Worku, Mulugeta Aemero.

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
