## [Decision Letter · Decision Letter 0]

20 Apr 2026

PONE-D-26-04595Species Composition, Deltamethrin Susceptibility, and kdr Mutations in Anopheles Mosquitoes across Two Transmission Seasons in Northwest EthiopiaPLOS One

Dear Dr. Worku,

Thank you for submitting your manuscript to PLOS ONE. After careful consideration, we feel that it has merit but does not fully meet PLOS ONE’s publication criteria as it currently stands. Therefore, we invite you to submit a revised version of the manuscript that addresses the points raised during the review process.

We look forward to receiving your revised manuscript.

Kind regards,

Michael E. von Fricken, MPH, PhD

Academic Editor

PLOS One

**Journal Requirements:**

1. When submitting your revision, we need you to address these additional requirements. Please ensure that your manuscript meets PLOS ONE's style requirements, including those for file naming. The PLOS ONE style templates can be found at https://journals.plos.org/plosone/s/file?id=wjVg/PLOSOne_formatting_sample_main_body.pdf and https://journals.plos.org/plosone/s/file?id=ba62/PLOSOne_formatting_sample_title_authors_affiliations.pdf 2. In your Methods section, please provide additional information regarding the permits you obtained for the work. Please ensure you have included the full name of the authority that approved the field site access and, if no permits were required, a brief statement explaining why. 3. Your ethics statement should only appear in the Methods section of your manuscript. If your ethics statement is written in any section besides the Methods, please move it to the Methods section and delete it from any other section. Please ensure that your ethics statement is included in your manuscript, as the ethics statement entered into the online submission form will not be published alongside your manuscript. 4. We note that Figure 1 in your submission contain map images which may be copyrighted. All PLOS content is published under the Creative Commons Attribution License (CC BY 4.0), which means that the manuscript, images, and Supporting Information files will be freely available online, and any third party is permitted to access, download, copy, distribute, and use these materials in any way, even commercially, with proper attribution. For these reasons, we cannot publish previously copyrighted maps or satellite images created using proprietary data, such as Google software (Google Maps, Street View, and Earth). For more information, see our copyright guidelines: http://journals.plos.org/plosone/s/licenses-and-copyright. We require you to either present written permission from the copyright holder to publish these figures specifically under the CC BY 4.0 license, or remove the figures from your submission: a. You may seek permission from the original copyright holder of Figure 1 to publish the content specifically under the CC BY 4.0 license.   We recommend that you contact the original copyright holder with the Content Permission Form (http://journals.plos.org/plosone/s/file?id=7c09/content-permission-form.pdf) and the following text:“I request permission for the open-access journal PLOS ONE to publish XXX under the Creative Commons Attribution License (CCAL) CC BY 4.0 (http://creativecommons.org/licenses/by/4.0/). Please be aware that this license allows unrestricted use and distribution, even commercially, by third parties. Please reply and provide explicit written permission to publish XXX under a CC BY license and complete the attached form.” Please upload the completed Content Permission Form or other proof of granted permissions as an "Other" file with your submission. In the figure caption of the copyrighted figure, please include the following text: “Reprinted from [ref] under a CC BY license, with permission from [name of publisher], original copyright [original copyright year].” b. If you are unable to obtain permission from the original copyright holder to publish these figures under the CC BY 4.0 license or if the copyright holder’s requirements are incompatible with the CC BY 4.0 license, please either i) remove the figure or ii) supply a replacement figure that complies with the CC BY 4.0 license. Please check copyright information on all replacement figures and update the figure caption with source information. If applicable, please specify in the figure caption text when a figure is similar but not identical to the original image and is therefore for illustrative purposes only.The following resources for replacing copyrighted map figures may be helpful: USGS National Map Viewer (public domain): http://viewer.nationalmap.gov/viewer/The Gateway to Astronaut Photography of Earth (public domain): http://eol.jsc.nasa.gov/sseop/clickmap/Maps at the CIA (public domain): https://www.cia.gov/library/publications/the-world-factbook/index.html and https://www.cia.gov/library/publications/cia-maps-publications/index.htmlNASA Earth Observatory (public domain): http://earthobservatory.nasa.gov/Landsat:
http://landsat.visibleearth.nasa.gov/USGS EROS (Earth Resources Observatory and Science (EROS) Center) (public domain): http://eros.usgs.gov/#Natural Earth (public domain): http://www.naturalearthdata.com/ 5. We note that Figures S2, S3 and S4 in your submission contain copyrighted images. All PLOS content is published under the Creative Commons Attribution License (CC BY 4.0), which means that the manuscript, images, and Supporting Information files will be freely available online, and any third party is permitted to access, download, copy, distribute, and use these materials in any way, even commercially, with proper attribution. For more information, see our copyright guidelines: http://journals.plos.org/plosone/s/licenses-and-copyright. We require you to either present written permission from the copyright holder to publish these figures specifically under the CC BY 4.0 license, or remove the figures from your submission: a. You may seek permission from the original copyright holder of Figures S2, S3 and S4  to publish the content specifically under the CC BY 4.0 license.  We recommend that you contact the original copyright holder with the Content Permission Form (http://journals.plos.org/plosone/s/file?id=7c09/content-permission-form.pdf) and the following text:“I request permission for the open-access journal PLOS ONE to publish XXX under the Creative Commons Attribution License (CCAL) CC BY 4.0 (http://creativecommons.org/licenses/by/4.0/). Please be aware that this license allows unrestricted use and distribution, even commercially, by third parties. Please reply and provide explicit written permission to publish XXX under a CC BY license and complete the attached form.” Please upload the completed Content Permission Form or other proof of granted permissions as an "Other" file with your submission.  In the figure caption of the copyrighted figure, please include the following text: “Reprinted from [ref] under a CC BY license, with permission from [name of publisher], original copyright [original copyright year].” b. If you are unable to obtain permission from the original copyright holder to publish these figures under the CC BY 4.0 license or if the copyright holder’s requirements are incompatible with the CC BY 4.0 license, please either i) remove the figure or ii) supply a replacement figure that complies with the CC BY 4.0 license. Please check copyright information on all replacement figures and update the figure caption with source information. If applicable, please specify in the figure caption text when a figure is similar but not identical to the original image and is therefore for illustrative purposes only. 6. We note that Figure S1 includes an image of a participant in the study. As per the PLOS ONE policy (http://journals.plos.org/plosone/s/submission-guidelines#loc-human-subjects-research) on papers that include identifying, or potentially identifying, information, the individual(s) or parent(s)/guardian(s) must be informed of the terms of the PLOS open-access (CC-BY) license and provide specific permission for publication of these details under the terms of this license. Please download the Consent Form for Publication in a PLOS Journal (http://journals.plos.org/plosone/s/file?id=8ce6/plos-consent-form-english.pdf). The signed consent form should not be submitted with the manuscript, but should be securely filed in the individual's case notes. Please amend the methods section and ethics statement of the manuscript to explicitly state that the patient/participant has provided consent for publication: “The individual in this manuscript has given written informed consent (as outlined in PLOS consent form) to publish these case details”.  If you are unable to obtain consent from the subject of the photograph, you will need to remove the figure and any other textual identifying information or case descriptions for this individual. 7. Please include captions for your Supporting Information files at the end of your manuscript, and update any in-text citations to match accordingly. Please see our Supporting Information guidelines for more information: http://journals.plos.org/plosone/s/supporting-information. 8. If the reviewer comments include a recommendation to cite specific previously published works, please review and evaluate these publications to determine whether they are relevant and should be cited. There is no requirement to cite these works unless the editor has indicated otherwise.

**Additional Editor Comments:**

Please address reviewer one comments.

Reviewers' comments:

Reviewer's Responses to Questions

**Comments to the Author**

1. Is the manuscript technically sound, and do the data support the conclusions?

Reviewer #1: Yes

2. Has the statistical analysis been performed appropriately and rigorously? 

Reviewer #1: Yes

3. Have the authors made all data underlying the findings in their manuscript fully available?

Reviewer #1: Yes

4. Is the manuscript presented in an intelligible fashion and written in standard English?

Reviewer #1: No

5. Review Comments to the Author

**Reviewer #1:** reviewer comments

Manuscript title: Species Composition, Deltamethrin Susceptibility, and kdr Mutations in Anopheles Mosquitoes across Two Transmission Seasons in Northwest Ethiopia

I. Main comments

1. The title of the manuscript needs to be reformulated to adhere the main objective.

2. In the abstract section, methodology and results need to be clarified. As the strain used (field and lab strain) or only field strain. Number of mosquitoes sample used for PCR. They were sampling from the exposed to deltamethrin mosquitoes or what? Idem for results.

II. Minor comments

Introduction section:

Line 61: can survive exposure to WHO diagnostic dose

Line 85-86: “This study seeks to provide comprehensive assessment of Anopheles mosquito populations in Maksegnit and Gendawuha towns of Northwest 86 Ethiopia.” Please rephrase to take into account all the main words of the tittle.

Line 87: seasonal and geographical variation of what?

Methods and materials

Line 106: Mosquito larvae, pupa collection, rearing, and adult collection, please specified clearly that mosquitoes were collected during the two saisons (rainy and after rainy)

Line 110: replace grown by reared

Line 111-112: reformulate the phrase to make easy to understand. Please give the quantity of Tetramin fish per tray and approximatively the number of larvae per tray.

Line 112: rephrase “Pupae were transferred into adult emergence cages and allowed to emerge into adults” by Pupae were collected each morning with…. and transferred into adult emergence cages to allow emerge into adults

Line 121: rephrase “Twenty-five 3–5-day-old female Anopheles mosquitoes” into Twenty-five 3–5-day-old female Anopheles mosquitoes from field larvae collection,

Line 122: deltamethrin-impregnated papers at what concentration?

Line 123: Control groups were left unexposed: control group must be exposed to untreated paper in the same condition as treated

Line 123: After a one-hour exposure replace by After one-hour of exposure

Line 125: When control mortality was between 5% and 20%, results were corrected using Abbott's formula. It was the case? If yes rephrase by treated mortality where corrected using Abbott's formula as control mortality was between 5% and 20%

Line 129: 90–129 97% mortality indicates a resistance candidate please check and rephrase

Line 131: DNA extraction and species identification, please give the equipment and serial number used here

Line 157: Detection of the L1014F and L1014S kdr alleles, please give the equipment and serial number used here

Results

Line 175: correct result by results

Line 177: and replace by in the

Line 180: No seasonal variation was observed in the distribution of species, please give the statical result as p-value

Line 182: groups (Fig. 2B-D), it shows two groups of mosquitoes and in line 177 you give only adult collection number. What about the exposure to deltamethrin samples used?

Line 190: delate during

Line 193: can you review the table tittle?

Line 201 – 204: can you give the number of each sub-group?

Line 218: please be consistent with the expression. Used phenotypically susceptible instead of susceptible. Harmonize in the test.

Figures section

Fig 2: (A) shows the frequency of mosquito species (B) the species composition frequency by

season and site for field collected adult (C), laboratory-reared susceptible (D), and laboratoryreared

resistant mosquitoes. Replace by Fig 2: (A) Prevalence of mosquito species (B) Species composition frequency by season and site for field collected adult (C) phenotypical susceptible mosquitoes after exposure to deltamethrin species frequency (D) phenotypical resistant mosquitoes after exposure to deltamethrin species frequency.

Fig 3: develop the tittle to allow well comprehensive and also the legend

6. PLOS authors have the option to publish the peer review history of their article (what does this mean?). If published, this will include your full peer review and any attached files.

Reviewer #1: No

---

## [Author Response · Author response to Decision Letter 1]

9 May 2026

We have revised the Methods section to include detailed information regarding the permits obtained for this study. Specifically, we have now clearly stated the full name of the authority that approved access to the field sites (line 118-119).

---

## [Editor Report · Decision Letter 1]

20 May 2026

Seasonal Variation in Species Composition, Deltamethrin Susceptibility, and kdr Mutations in Anopheles Mosquitoes in Northwest Ethiopia

PONE-D-26-04595R1

Dear Dr. Worku,

We’re pleased to inform you that your manuscript has been judged scientifically suitable for publication and will be formally accepted for publication once it meets all outstanding technical requirements.

Kind regards,

Michael E. von Fricken, MPH, PhD

Academic Editor

PLOS One
---

## [Editor Report · Acceptance letter]

PONE-D-26-04595R1

PLOS One

Dear Dr. Worku,

I'm pleased to inform you that your manuscript has been deemed suitable for publication in PLOS One. Congratulations! Your manuscript is now being handed over to our production team.

Kind regards,

on behalf of

Dr. Michael E. von Fricken

Academic Editor

PLOS One